# Examining the Effect of Self-Determined Appeal Organ Donation Messages and Respective Underlying Mechanism

**DOI:** 10.3390/ijerph191710619

**Published:** 2022-08-25

**Authors:** Sining Kong

**Affiliations:** Department of Communication and Media, College of Liberal Arts, Texas A&M University at Corpus Christi, Corpus Christi, TX 78412, USA; sining.kong@tamucc.edu

**Keywords:** organ donation, self-determination theory, self-integrity, sympathy, pride

## Abstract

This study examined how intrinsic motivation and its respective underlying mechanism influence people’s attitude and intentions of organ donation. The findings revealed the importance of meeting people’s customized psychological needs. For the general population, especially non-organ donors, autonomous appeal message will be more effective in promoting their intention of becoming an organ donor. For registered organ donors, competence-based organ donation messages are more effective in increasing their promotion and seeking behavior of organ donation. This study also discovered underlying mechanisms of intrinsic motivation, such as self-integrity, pride, and sympathy. Pairing underlying mechanism with competence-based messages can maximize the message impact.

## 1. Introduction

The shortage of organ transplantation is a persisting problem in United States. Although over 41,000 transplants were performed in 2021, there were still 106,698 of men, women, and children on the national transplant waiting list [1]. Although 90% of the US population supports organ donation, only 60% are registered as organ donors. The discrepancy between attitude and behavior change reflects a dilemma in health communication. Persuading people to become organ donors is one of the health-related behaviors. In addition to behavior change in organ donation, prior studies also examined other health-related behaviors change in various areas. For example, Latkin and Knowlton [2] investigated how extrinsic motivation, such as social norms and social rewards, influences health behavior change. Silva et al. [3] studied how intrinsic motivation, such as self-determination theory-based interventions, affects health behavior change. Although previous studies in health communication examined both extrinsic and intrinsic motivations in behavior change, when it comes to behavior change in organ donation, extrinsic motivations have been extensively explored. Previous studies on organ donation mainly focused on examining extrinsic motivation, such as avoiding guilt [4] or emphasizing self-benefit [5], the role of altruism in organ donation [6], and moral emotions, such as sympathy and pride [7] to persuade people to become organ donors. The methods used by previous studies were mainly experimental design [4] and survey [5]. They reached the conclusion via convenience sample by either using undergraduate students or Mturk workers. Few studies have examined how intrinsic motivations affect organ donation. To resolve the research gap, this study explores how intrinsic motivations [8] and the respective underlying mechanism affect organ donation.

As intrinsic motivations can help people regain self-integrity by reassuring people’s core values, emphasizing interpersonal connection, and increasing self-adequacy [9,10], self-integrity is a potential mediator. According to Dillard and Shen [11], threat to freedom is one of the major reasons that public health campaigns fail to produce desired results. Intrinsic motivations, such as autonomy and competence, allow people to have freedom to choose and perform certain behavior through reduced perceived threat to freedom. Hence, threat to freedom is another potential mediator. Furthermore, as sympathy values interdependent relationships which is consistent with relatedness, and pride emphasizes personal achievement which corresponds to competence, these two moral emotions can also be viewed as potential mediators. Thus, in addition to examining how intrinsic motivations influence people’s organ donation attitude and intentions, this study also explores how the underlying mechanisms, such as self-integrity, perceived threat to freedom, sympathy, and pride, respectively mediate the effect between self-determined appeal messages and attitude and intentions of organ donation.

## 2. Literature Review

### 2.1. Self-Determination Theory

Self-determination theory contends that autonomy, competence, and relatedness are people’s basic psychological needs [8]. Autonomy reflects whether a person’s behavior is volitional and emanating from one’s personal values. Relatedness reflects a feeling of connection and belonging to others, rather than isolation or loneliness. Competence reflects whether a person feels capable of achieving desired behavior [12]. When these basic psychological needs are satisfied, people are more likely to perform prosocial behavior.

Autonomy as a volitional and motivational state emphasizes behavior, which is congruent with one’s self and has an internal locus of control for certain behavior [13]. Autonomous prosocial motivation reflects a person’s intrinsic desire to help [14] and leads to greater well-being in the helper than controlled prosocial motivation [15]. Pavey et al. [16] found that controlled prosocial behavior had no positive effects in helping through a university student-based experimental design. Because controlled prosocial behavior is not consistent with people’s goals and values, it tends to generate backlash. In contrast, autonomous motivated prosocial behavior is free from external duty or fear of negative consequences for not performing the behavior. Therefore, autonomous appeal organ donation message will be more effective than controlled organ donation message in generating prosocial behavior.

Relatedness refers to a feeling of connection with others [17]. Relatedness perceives helping as an interpersonal behavior and emphasizes cohesiveness or intimacy [15]. This argument indicates that from the relatedness perspective, a mutually rewarding relationship is essential in prosocial behavior [18]. As Moller et al. [19] found that satisfaction of the need for relatedness makes people cherish relatedness experience further via a university student based experimental design, a fulfillment of relatedness will lead to more prosocial tendencies. Previous study revealed that both exposed to implicit relatedness priming task and asking participants to write their past relatedness experience can induce more prosocial activities than neutral condition [16]. Hence, a relatedness appeal organ donation message will be more effective than a controlled organ donation message in generating prosocial behavior.

Competence is defined as “feeling effective in one’s ongoing interactions with the social environment and experiencing opportunities to exercise and express one’s capacities” [12]. Instead of focusing on an attained skill, competence emphasizes a sense of confidence and capability in action. When the need for competence is fulfilled, it will enhance people’s intrinsic motivation and lead to behavior change. Bachmann et al. [20] conducted a patient-based survey and found that an increased competence was positively associated with people’s health related behavior change. Thus, improved competence in organ donation will empower people and motivate them to become organ donors.

Furthermore, previous studies examined how different types of message framing affect people’s attitude and intention of organ donation. For example, Reinhart et al. [4] found that gain-framed messages produced more positive reactions of organ donation through a series of experimental design. Chien and Chang [21] conducted an experiment and discovered that a loss-framed exemplar message generated more positive intentions toward organ donation than loss-framed statistical messages. Moreover, Sun [22] also revealed that positively framed message elicited more positive attitude towards organ donation and emotional appeal message resulted in a higher intention of becoming an organ donor via an experimental design. In the similar vein, this study focuses on how self-determined appeal organ donation messages influence people’s attitude and intention of organ donation.

**H1.** 
*Self-determined appeal organ donation messages will generate **(a)** more positive attitude, **(b)** more intention of becoming an organ donor, **(c)** more intention of promoting organ donation, and **(d)** more intention of seeking organ donation related information than controlled message.*


### 2.2. Self-Integrity

Self-integrity reflects a sense of global efficacy and indicates whether a person can behave morally or adaptively. The motive for self-integrity is to be a good moral and adaptive person. Threats to that self-integrity related image trigger psychological threat [23,24]. Threatening cognitions of self-integrity can derive from various sources, such as from the information in the environment, from others’ behavior toward us, from others’ judgment, from our behaviors, and from our cognitive response to certain situations [23]. In the context of organ donation, people’s self-integrity, which is an overall moral evaluation of oneself, is threatened by the cognitive response of organ donation. People either refer to their core personal values, or spend time with friends, or reflect on their values to regain self-integrity [9,10]. McQueen and Klein [9] conducted a systematic review of 47 eligible articles regarding experimental manipulations of self-affirmation. Any affirmation of important aspects of the self helps regain self-integrity. Napper et al. [10] adopted values in action (VIA) strengths scale in an experimental design to elicit students’ positive appraisal of themselves to improve their self-integrity. These copying strategies correspond with autonomy, relatedness, and competence, respectively. As organ donation is an ambivalent prosocial behavior [25], the tug of war between helping others and the concern of mortality threaten people’s overall self-integrity. Therefore, intrinsic motivations can help people regain self-integrity by reassuring people’s core values (e.g., autonomy), emphasizing interpersonal connection (e.g., relatedness), and increasing self-adequacy (e.g., competence).

**H2.** 
*Self-integrity will mediate the effect between all self-determined messages and attitude and intentions of organ donation.*


### 2.3. Perceived Threat to Freedom

According to psychological reactance theory, people value the ability to choose among alternatives. Whenever the freedom to choose is threatened or eliminated, state reactance occurs [26]. In psychological reactance theory, freedom is “not abstract considerations, but concrete behavioral realities” [27]. Both knowledge and ability are integral components of a free behavior. When people possess enough knowledge and ability to perform certain behavior and can choose among alternatives, they can retain their freedom.

Different types of messages that reflect varying extent of people’s knowledge, capability, and ability to choose alternatives can help people restore their freedom to different extents. Smit et al. [28] conducted an experiment by recruiting 526 participants via a research panel and revealed that autonomy-supportive language could improve a sense of freedom and elicited more positive evaluation of the intervention than controlled language. Smith et al. [29] also discovered that a strong sense of self-efficacy was negatively associated with perceived threat to freedom by conducting a longitudinal study of 402 women offenders on probation or parole. In the context of organ donation, perceived threat to freedom derives from a deprivation of choosing the alternative option of not becoming an organ donor, and a lack of knowledge of what they can achieve by becoming an organ donor. In this study, an autonomous appeal message provides people alternatives by convincing them that they have a choice of either becoming an organ donor or not. A competent appeal message improves both people’s knowledge by resolving people’s concerns of organ donation, and people’s capability of organ donation by informing them how many lives they can save. Hence, both autonomy and competence will reduce perceived threat to freedom and generate more positive attitude and intentions of organ donation.

**H3.** 
*Perceived threat to freedom will mediate the effect between **(a)** autonomous appeal message, **(b)** competent appeal message, **(c)** a combined autonomous and competent appeal message, and **(d)** a combined related and competent appeal message and attitude and intentions of organ donation.*


### 2.4. Sympathy

Sympathy is an other-focused emotion and reflects interdependent relationships with others [30,31]. Victims’ group belonging and identity visibility affect people’s sympathy and prosocial tendencies. People tend to help those whom they perceive as in-group members and similar to themselves rather than out-group members [32]. The in-group connection gives people a sense of “weness”, which promotes prosocial behavior [33]. Furthermore, previous study also found that identifiable victims trigger more sympathy than statistical victims [34]. Identifiable victims are more vivid than statistical victims and people tend to pay more cognitive attention to them [35]. In the context of organ donation, sympathy reflects if people can understand the suffering and feeling of people who need an organ to survive. Related appeal messages which emphasize connection with other people and saves our loved ones’ lives reflects both group belonging and victims’ identities. Thus, a related appeal organ donation message will generate more sympathy than unrelated message. Moreover, sympathy is also a moral emotion [36] and can positively influence organ donation attitude [37] and intention [7,38]. Massi Lindsey and Yun [37] found that sympathy positively associated with attitude via an experiment. Kong and Lee [7] and Kong [38] conducted an experiment on Mturk and found that sympathy exerted a significant impact on attitude and intention of organ donation. Therefore, under a related appeal organ donation message, people tend to experience more sympathy, which leads to a more positive attitude and intention of organ donation.

**H4.** 
*Sympathy will mediate the effect between **(a)** related appeal message and **(b)** a combined related and competent appeal message and attitude and intentions of organ donation.*


### 2.5. Pride

In contrast, pride is a self-conscious emotion deriving from personal achievement [39]. Pride is intrinsically linked to self-esteem and many acts of self-improvement can promote pride. Tracy and Robin [39] also proposed two facets of pride: authentic and hubristic. Authentic pride highlights internal attributes and unstable, controllable causes; for example: “I performed well because I practiced.” In contrast, hubristic pride not only reflects internal attributes, but also emphasizes stable, uncontrollable causes (e.g., “I performed well because I am always great”). Given the features of the two facets of pride, authentic pride—formed by specific behavior—wins over hubristic pride in terms of prosocial behavior. Since pride is also a moral emotion [40], feelings of pride can produce more prosocial behavior [41]. Kong and Lee [7] found that pride appeal message generated more positive attitude and intention of organ donation via an experiment with Amazon Mturk workers. In the context of organ donation, pride denotes if a person can do anything to help others who are suffering. Hence, competence which highlights people’s confidence and capacity corresponds to the characteristics of pride. Competent appeal organ donation message which focuses on improving people’s self-efficacy of becoming organ donors can generate more pride, which will lead to more positive attitude and intention of organ donation.

**H5.** 
*Pride will mediate the effect between **(a)** competent appeal message, **(b)** a combined autonomous and competent appeal message, and **(c)** a combined related and competent appeal message and attitude and intentions of organ donation.*


Given the feature of autonomy, relatedness, and competence, this study also proposed a series of parallel mediation effects for the combined autonomous/related and competent appeal message.

**H6.** 
*Self-integrity, pride, and perceived threat to freedom will respectively mediate the effect between a combined autonomous and competent appeal message and attitude and intentions of organ donation.*


**H7.** 
*Self-integrity, sympathy, pride, and perceived threat to freedom will respectively mediate the effect between a combined related and competent appeal message and attitude and intention of organ donation.*


**RQ1.** 
*How does organ donor status moderate the hypothesized relationship?*


## 3. Method

### 3.1. Experimental Design and Procedure

An online experiment was conducted to test the hypotheses of this study. There were six conditions of the experiment: (1) autonomous appeal message, (2) related appeal message, (3) competent appeal message, (4) a combined autonomous and competent appeal message, (5) a combined related and competent appeal message, and (6) controlled message. Participants were randomly assigned to one of the conditions. After reading the assigned message, participants were asked to fill out a survey of their attitudes, intentions of organ donation, self-integrity, perceived threat of freedom, sympathy, and pride.

### 3.2. Participants

A total of 364 participants who lived in the US and had a 95% approval rate were recruited from Amazon Mechanical Turk. After deleting three incomplete questionnaires, 361 valid participants were retained. There were 234 male participants (64.8%), 123 female participants (34.1%), and 4 participants self-identified as other gender (1.1%). A total of 284 participants were Caucasians (78.7%), 31 participants were African American (8.6%), 22 were Latino/Hispanic (6.1%), 19 Asian/Asian American (5.3%), and 15 were American Indian or Alaska Native (4.2%). Among them, 3 participants had less than high school degree (0.8%), 28 participants had high school diploma (7.8%), 26 participants had some college but no degree (7.2%), 16 participants had associate degree in college (4.4%), 191 participants had a bachelor’s degree (52.9%), 91 participants had a master’s degree (25.2%), 4 participants had a doctoral degree (1.1%), and 2 participants had a professional degree.

### 3.3. Stimuli

To ensure the ecological validity of this study, the controlled message and competent appeal message of this study were adapted from two major organ donation websites: United Network for Organ Sharing and Organdonor.gov to persuade people to become organ donors. As both websites include current organ donation situation (e.g., listing how many people on the national organ transplant waiting list), that common information was adopted as a controlled organ donation message in this study. Competent appeal messages were adapted from both United Network for Organ Donation (e.g., organ donation is not against religions), and Organdonor.gov (e.g., every donor can save 8 lives and enhance over 75 more).

Autonomous appeal message was manipulated to reflect people’s core values by admitting their concerns of organ donation (e.g., it is a tough decision to make because of religious belief, cultural values, and medical mistrust), freeing people from the fear of not being an organ donor (e.g., although you are not an organ donor, you will still be a good person), and emphasizing people’s free choice (e.g., whether to be an organ donor or not is entirely up to you). Related appeal message was manipulated to focus on the connection with others (e.g., organ donation is not saving nobodies’ lives, but to save our loved ones’ lives).

### 3.4. Pretest

A pretest (N = 179) was conducted to examine the efficacy of the manipulated message. Planned contrast results showed that autonomous appeal message (*M* = 5.66, *SD* = 1.25) generated a marginally higher level of autonomy than controlled message (*M* = 5.02, *SD* = 1.32), t (173) = 2.13, *p* = 0.034. The planned contrast results also showed the combined autonomous and competent appeal message (*M* = 5.98, *SD* = 1.14) generated significantly higher level of autonomy than controlled message (*M* = 5.02, *SD* = 1.32), t (173) = 3.21, *p* = 0.002. Hence, autonomous associated appeal message was successfully manipulated. As for relatedness, planned contrast results showed that related appeal message generated higher level of relatedness (*M* = 6.10, *SD* = 0.96) than controlled message (*M* = 5.18, *SD* = 1.30), t (173) = 2.59, *p* = 0.010. When it comes to competence, planned contrast results showed that competent appeal message generated higher level of competence (*M* = 5.88, *SD* = 1.12) than controlled message (*M* = 5.35, *SE* = 1.05), t (173) = 1.97, *p* = 0.050. Therefore, competent appeal message is also successfully manipulated.

### 3.5. Measurement

Dependent variables. Attitudes toward organ donation were measured with six items adapted from Feeley and Servoss [42] (Cronbach’s alpha = 0.86). Intentions to register or continue one’s status as an organ donor were measured with three items adopted from Wang and Zhao [43] (α = 0.91). Intentions of promoting organ donation were measured with four items adopted from Chun and Lee [44] (α = 0.91). Intentions of seeking more information were measured with three items adopted from Wang and Zhao [43] (α = 0.92). All dependent variables were measured on a 7-point Likert scale (1 = Strongly disagree to 7 = Strongly agree).

Mediating variables. Self-integrity was measured with four items adapted from Sherman et al. [45] (Cronbach’s alpha = 0.78) (e.g., I feel that I’m basically a moral person; I try to do the right thing). Perceived threat to freedom was measured with four items adopted from Dillard and Shen [11] (α = 0.92) (e.g., the message threatened my freedom to choose; the message tried to make a decision for me). Sympathy was measured with three items adapted from Vossen, Piotrowski, and Valkenburg [46] (α = 0.64) (e.g., I feel sorry for people who needs organ transplantation; I feel concerned for people who are suffering from organ failure). Pride was measured with three items adapted from Krettenauer and Casey [47] (α = 0.87) (e.g., I feel I can help people who are suffering from organ failures; I feel I can do something good to help people who need organ transplantation). All the mediating variables were measured on a 7-point Likert scale (1 = Strongly disagree to 7 = Strongly agree) and demonstrated an acceptable to excellent reliability.

## 4. Results

The data were analyzed with ANCOVA (analysis of covariance) and SEM (structural equation model). ANCOVA was used to examine the effect among different types of organ donation messages on attitude and intentions of organ donation. SEM was used to examine the relationship among latent variables via MPlus 8. According to Hu and Bentler [48], the RMSEA with a value of 0.06 or lower and the SRMR with a value of 0.08 or lower indicated a good model fit. Furthermore, Kline [49] claimed that CFI values greater than 0.90 may indicate a reasonably good fit. Prior to hypothesis testing, CFA was conducted to analyze the measurement items for the proposed model. The results showed that most indicators of latent variables had standardized factor loadings higher than 0.50 and significant at 0.001 level, which affirms the convergent validity of the measurement [50]. Average variance extracted (AVE) was greater than 0.05 and composite reliability (CR) was greater than 0.70. Therefore, the results indicated adequate convergent validity, discriminant validity, and reliability of the measurement model. For the following analyses, the effects of gender, age, and donor status were all statistically controlled.

### Hypotheses Testing

H1 asserted that self-determined appeal message would be more effective than controlled message in generating more positive attitude and intentions of organ donation. A series of ANCOVA were conducted. The results showed that there was no significant effect among different types of organ donation messages on attitude and intentions of organ donation (*p* > 0.05). Although the pairwise comparisons results showed that autonomous appeal message (*M* = 5.81, *SD* = 1.12) generated more intentions of becoming organ donors than a combined related and competent appeal message (*M* = 5.19, *SD* = 1.82) in a marginally significant level, *p* = 0.064, there was no significant difference between self-determined appeal messages and controlled message in other dependent variables. H1 was not supported.

H2 asserted that self-integrity would mediate the effect between self-determined messages and attitude and intentions of organ donation. Results showed that self-integrity significantly mediated the effect between competent appeal message and attitude towards organ donation (Indirect Effect (IE) = 0.22, 95%CI = 0.096, 0.330), intentions of becoming organ donors (IE = 0.26, 95%CI = 0.107, 0.411), promoting organ donation (IE = 0.18, 95%CI = 0.070, 0.298) and seeking organ donation-related information (IE = 0.19, 95%CI = 0.077, 0.305) (see Figure 1). Self-integrity also significantly mediated the effect between a combined related and competent appeal message and attitudes toward organ donation (IE = 0.21, 95%CI = 0.080, 0.367), intention of becoming organ donors (IE = 0.25, 95%CI = 0.097, 0.427), promoting organ donation (IE = 0.17, 95%CI = 0.062, 0.290), and seeking organ donation-related information (IE = 0.18, 95%CI = 0.058, 0.325) (see Figure 2). Therefore, H2 was partially supported.

H3 asserted that perceived threat to freedom would mediate the effect between autonomous-related and competent-related message. Results showed that perceived threat to freedom did not significantly mediate the effect between autonomous, competent, and a combination of related and competent appeal message and attitude and intentions of organ donation (*p* > *0*.05). Perceived threat to freedom only significantly mediated the effect between the combined autonomous and competent appeal message and attitude towards organ donation (IE = 0.05, 95%CI = 0.010, 0.118), intention of promoting organ donation (IE = −0.13, 95%CI = −0.270, −0.040) and seeking organ donation-related information (IE = −0.14, 95%CI = −0.270, −0.040) (see Figure 3). Although the reduced perceived threat to freedom led to more positive attitude towards organ donation, it generated less intention of promoting organ donation and seeking organ donation-related information. Therefore, H3 was not supported.

H4 asserted that sympathy would mediate the effect between messages which reflected relatedness and attitude and intentions of organ donation. Results showed that sympathy only significantly mediated the effect between a combined related and competent appeal message and attitude (IE = 0.27, 95%CI = 0.077, 0.471), intention of becoming organ donors (IE = 0.28, 95%CI = 0.092, 0.480), promoting organ donation (IE = 0.11, 95%CI = 0.011, 0.233), and seeking organ donation-related information (IE = 0.13, 95%CI = 0.016, 0.259). Therefore, H4 was partially supported (see Figure 4). Furthermore, sympathy also significantly mediated the effect between competent appeal message and all dependent variables, and significantly mediated the effect between the combined autonomous and competent appeal message and attitude and intention of becoming organ donors.

H5 asserted that pride would mediate the effect between messages which reflected competence and attitude and intentions of organ donation. Results showed that pride only significantly mediated the effect between competent appeal message and attitude towards organ donation (IE = 0.09, 95%CI = 0.016, 0.213), intentions of becoming organ donors (IE = 0.23, 95%CI = 0.049, 0.427), promoting organ donation (IE = 0.24, 95%CI = 0.057, 0.432), and seeking organ donation-related information (IE = 0.26, 95%CI = 0.056, 0.470) (see Figure 5). Therefore, H5 was partially supported.

Both H6 and H7 predicted a series of parallel mediation effects between self-determined appeal message and attitude and intentions of organ donation. The results showed that there were neither significant mediation effects (*p* > 0.05) nor good model fit for H6 and H7. Therefore, both H6 and H7 were not supported.

The research question asked how the organ donor status moderated the hypothesized relationship. For the effect of self-determined appeal message on dependent variables, ANCOVA results showed that for both organ donors and non-organ donors, there were no significant differences among all types of messages regarding attitude towards organ donation. However, for non-organ donors, autonomous appeal message (*M* = 5.44, *SE* = 1.34) generated significantly higher level of intention of becoming organ donors than competent appeal message (*M* = 4.49, *SE* = 1.74), *p* = 0.044, a combined related and competent appeal message (*M* = 4.14, *SE* = 2.15), *p* = 0.006, and controlled message (*M* = 4.43, *SE* = 1.72), *p* = 0.033. Furthermore, related appeal message (*M* = 5.06, *SE* = 1.61) also generated significantly higher level of intention of becoming organ donors than a combined related and competent appeal message (*M* = 4.14, *SE* = 2.15) *p* = 0.032. Therefore, for non-organ donors, autonomous appeal message significantly generated more intention of becoming organ donors than other types of messages, and the combined related and competent appeal message generated least level of intention of becoming organ donors.

As for intention of promoting organ donation, the pairwise comparison results showed that for organ donors, controlled message (*M* = 5.31, *SD* = 1.38) was marginally more effective than related appeal message (*M* = 4.58, *SD* = 1.64) in producing intention of promoting organ donation, *p* = 0.067. A combined autonomous and competent appeal message (*M* = 5.21, *SD* = 1.19) was marginally more effective than related appeal message, *p* = 0.089. For non-organ donors, the pairwise comparison results showed that autonomous appeal message (*M* = 5.48, *SD* = 1.19) was significantly more effective than competent appeal message (*M* = 4.26, *SD* = 1.83), *p* = 0.027, and marginally more effective than the combined related and competent appeal message (*M* = 4.47, *SD* = 1.65), *p* = 0.072, in persuading people to promote organ donation.

For intention of seeking organ donation-related information, pairwise comparison results showed that for organ donors, the combined related and competent appeal message (*M* = 5.34, *SD* = 1.76) significantly generated more intention of seeking organ donation-related information than related appeal message (*M* = 4.56, *SD* = 1.68), *p* = 0.04. However, for non-organ donors, no significant findings were discovered. As for hypothesized mediation effect, the results showed no differences in the hypothesized mediation effect between organ donors and non-organ donors.

## 5. Discussion

This study examined the effectiveness of intrinsic motivations on people’s attitude and intentions of organ donation and the underlying mechanisms of the intrinsic motivations. The results indicated that when donor status, gender, and age were controlled, autonomous appeal message generated more intention of becoming organ donors than other types of messages. The effectiveness of autonomy can be explained by terror management theory. As organ donation reminds people of their own death, autonomy that acknowledges people’s concerns of organ donation and offers people’s free choice is consistent with people’s worldview value in organ donation. According to terror management theory, reminding people’s worldview value helps them cope with the fear of death [51].

Results also discovered that self-integrity, sympathy, and pride, in that order, significantly mediated the effect between competence-based message and attitude and intentions of organ donation. The mediation effect of self-integrity indicated that when bolstering people’s overall self-efficacy, people tend to perceive themselves as more capable and adaptive. This mediating role of self-integrity corresponds with previous studies that self-affirmed people are more open to threatening information [52,53]. The positive associations between moral emotions, such as sympathy and pride, and attitude and intentions of organ donation correspond with previous studies that both sympathy and pride can motivate people to become organ donors [7,37].

As for the perceived threat to freedom, results showed that when offering people freedom to choose, under the combined autonomous and competent appeal message, although a reduced perceived threat to freedom generated more positive attitude towards organ donation, it also significantly reduced people’s intention of promoting and seeking organ donation-related information. The negative association between perceived threat to freedom and attitude towards organ donation is consistent with Quick et al.’s [54] finding. However, the overall findings of perceived threat to freedom and series of parallel mediation effects contradict with anticipation. According to psychological reactance theory, when people’s freedom is threatened, one way to restore the freedom is to increase the liking of the threatened choice [11]. However, when people are offered the freedom to choose, they do not need to explore organ donation-related information to restore their freedom. That explains why reduced perceived threat to freedom diminished people’s intentions of promoting and seeking organ donation-related information. As for non-significant parallel mediation effect, it can be explained by VanderWeele and Vansteelandt’s [55] study which found that when including multiple mediating variables in a model, the mediators may affect one another and result in a failed effect.

To answer the research question, the results showed that although donor status did not moderate the hypothesized mediation effect, donor status moderated the effect between different types of messages and intentions of organ donation. For registered organ donors, both competence-based messages and controlled messages generated more intention of promoting and seeking organ donation-related information than related appeal message. According to terror management theory, either an enhanced self-esteem or reminding people of their own worldview value can help them cope with fear of death [56]. Competence-based organ donation messages which emphasized how many people a donor can save improve people’s self-esteem. The controlled message which described the current situation of organ donation reminds people of their worldview value as an organ donor, which is consistent with Jain and Ellithorpe’s [57] study. Hence, either competence-based message or controlled message will be more effective in persuading registered organ donors to promote and seek organ donation-related information.

As for non-organ donors, autonomous appeal message is the most effective message in changing people’s intention of becoming organ donors and promoting organ donation. As autonomy acknowledges people’s core values and releases people from the fear of aftermath of not performing prosocial behavior [14], autonomous appeal messages that endorse non-organ donors’ core values of organ donation strengthen their worldview value. The improved worldview value helps people cope with fear of death caused by organ donation [51]. As for the lack of effectiveness of competent and combined related and competent appeal message, that can be explained by the backfire triggered by competence which challenges non-organ donors’ beliefs and an exacerbated fear of death caused by related others. Thus, autonomous appeal organ donation message will be more effective in persuading non-organ donors to become registered donors and promote organ donation campaign than other types of messages, especially competent appeal, and a combined related and competent appeal message.

This study not only contributes to current organ donation study, but also expands self-determination theory. First, this study fills the research gap in examining how intrinsic motivations influence attitude and intentions of organ donation. Many studies in organ donation focused on applying extrinsic motivations in an organ donation campaign, such as providing incentives [58] or emphasizing self-benefit of organ donation [5]. This study adds to existential organ donation study by examining intrinsic motivations in organ donation. As organ donation can remind people of their own death [57], autonomy that admits people’s concerns of organ donation corresponds with people’s worldview value. The improved worldview value can be used as a copying mechanism to help people deal with fear of death [51]. Therefore, different from other prosocial behavior, when controlled for gender, age, and donor status, autonomy outperformed other types of intrinsic motivations in the context of organ donation.

Second, this study also contributes to organ donation study by revealing how donor status influences the effectiveness of different types of intrinsic motivations. This study discovered that registered organ donors value the psychological need of competence, while the non-organ donors cherish the psychological need of autonomy. This finding is consistent with prior studies that tailored message reflecting people’s beliefs can improve people’s behavioral outcomes [59,60]. Hence, understanding the features of registered and non-registered organ donors informs the importance of tapping into personalized intrinsic motivation in an organ donation campaign.

Third, this study enriched self-determination theory by discovering different underlying mechanisms of intrinsic motivations in organ donation. Different from other prosocial behavior, organ donation reflects altruistic motivation and reminds people of their own death at the same time [57,61]. To cope with the fear of death, pairing either self-integrity or pride with competence-based messages will improve people’s overall self-esteem, which helps people resolve the fear of death aroused by organ donation [56]. Given the altruistic feature of organ donation, pairing sympathy with competence-based messages will improve people’s altruistic motivation to become organ donors [37]. Therefore, this study discovered different causal sequences between self-determined appeal messages and attitude and intentions of organ donation.

This study has several practical implications that can advance the organ donation campaign. First, as organ donation reminds people of their own death, self-determined appeal messages which help people reduce fear of death will promote behavior change. The results showed that for both general populations and non-organ donors, autonomous appeal message was more effective in changing people’s intention of becoming organ donors. Hence, when targeting the general population, especially non-organ donors, practitioners should admit people’s concerns of organ donation, assure them that they will not be subjected to contempt for not being an organ donor, and offer them freedom of choice. The worldview value embodied in autonomy not only provides people social validation to defend against the fear of death, but also reduces psychological reactance.

Second, this study revealed the importance of creating personalized organ donation messages to meet both registered and non-registered donors’ needs. In contrast to non-organ donors who cherish autonomy, registered organ donors value competence and the worldview value of being an organ donor. To persuade registered organ donors to promote organ donation, practitioners can use a combined self-determined appeal message which is built on competence, such as a combined autonomous/related and competent appeal organ donation message, to meet registered donors’ multiple intrinsic motivations rooted in competence. To persuade registered organ donors to seek organ donation-related information, practitioners can merely describe the dire situation of organ donation to remind registered donors’ of the worldview value of being donors.

Third, this study also exhibited the value of incorporating mediated variables that correspond to intrinsic motivations into an organ donation campaign. Although current organ donation campaign includes competent appeal message to increase people’s self-efficacy and resolve people’s concerns and myths of organ donation, without highlighting the underlying mechanisms of competence-based messages, it is hard to change people’s behavioral intention of organ donation. Therefore, when targeting general population, organ donation campaign should either improve people’s overall self-adequacy of self-integrity or elicit pride or sympathy along with competence-based message to maximize the effect of organ donation messages.

To sum up, the finding of this study may revamp current organ donation campaign. As competent appeal message and controlled message in this study were adapted from two major organ donation websites, according to the results, current organ donation campaigns will be more effective in enhancing registered organ donors’ intention of promoting and seeking organ donation information. However, to improve the general population’s, especially non-organ donors’, intention of becoming organ donors and promoting organ donation, instead of using competent appeal message, applying autonomous appeal message in an organ donation campaign will be more effective.

There are several limitations of this study. First, given the nature of organ donation, it is hard to trigger single pure intrinsic motivation by using message manipulation. Future study could prime people with autonomy, relatedness, and competence to produce untainted intrinsic motivations. Second, although this study examined the effect of multiple intrinsic motivations based on current competent appeal organ donation messages, this study did not investigate the effect of combining all three intrinsic motivations nor did it examine the effect of combining autonomy and relatedness on organ donation. Future study could examine how meeting all of people’s psychological needs and people’s multiple psychological needs except competence affect people’s organ donation attitude and intention. Third, as organ donation can remind people of their own death, future study could examine the mediating role of self-esteem and worldview value of intrinsic motivations to help people cope with the fear of death caused by organ donation. Fourth, in addition to examining the persuasive effects of message framing, people’s daily experiences, such as education [62] and cultural background [63] also affect people’s organ donation attitude. Therefore, future study should also investigate how people’s daily experiences and demographic features, which include but are not limited to education and cultural background, moderate the persuasive effect of message framing in organ donation.

## 6. Conclusions

Over 95% people have a positive attitude of organ donation, but only 60% are signed up as organ donors in US. To resolve this problem, this study tapped into people’s intrinsic motivations to promote organ donation. According to the findings, autonomy that reflects people’s volition and core values is more effective in motivating general population, especially non-organ donors, to become organ donors. This study also reveals the importance of meeting people’s customized psychological needs. Organ donation campaigns should provide registered donors with competence-based messages to meet their multiple psychological needs which are centered on competence and non-organ donors with autonomous appeal messages to satisfy their psychological need of volition and an acknowledgement of their core values. This study also deciphers the effect of intrinsic motivation by revealing two causal sequences with different underlying mechanisms. One causal sequence is to reduce the fear of death via either self-integrity or pride. Highlighting either self-integrity or pride in competence-based messages could make the most of competence-based messages and generate a more positive attitude and intentions of organ donation. The other causal sequence is to stimulate people’s altruistic motivation via sympathy. Emphasizing sympathy in competence-based messages could elicit more positive attitude and intentions of organ donors through improved altruism. Therefore, understanding the effect of intrinsic motivations, distinguishing different types of organ donors’ psychological needs, and uncovering distinct underlying mechanisms of intrinsic motivations advance the knowledge of organ donation and expand self-determination theory.

## Figures and Tables

**Figure 1 ijerph-19-10619-f001:**
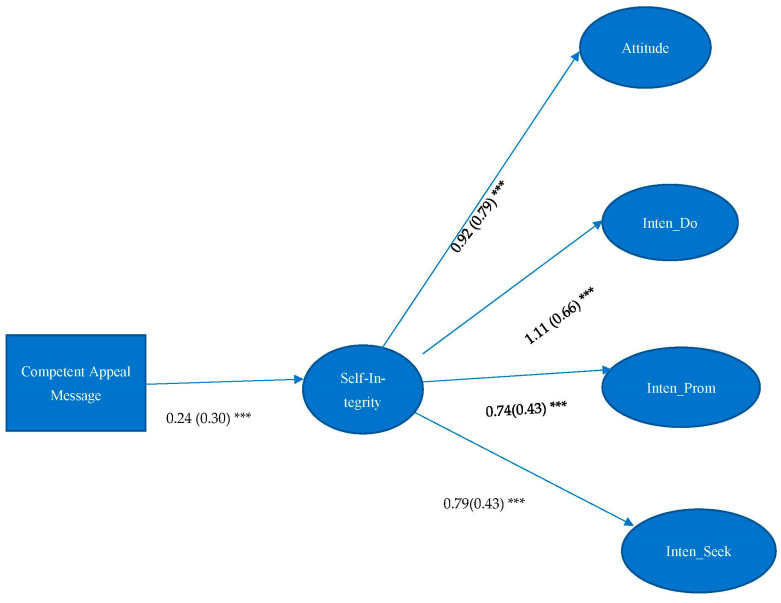
Unstandardized estimates are listed, followed by standardized estimates in parentheses. Model Fit: χ^2^ = 443.496, df = 224, *p* < 0.001; RMSEA < 0.001, 90% C.I.: 0.063–0.084, CFI = 0.920, SRMR = 0.054. Note: *** *p* < 0.001. Attitude = Attitude toward organ donation, Inten_Do = Intention of becoming organ donors, Inten_Prom = Intention of promoting organ donation, Inten_Seek = Intention of seeking organ donation-related information.

**Figure 2 ijerph-19-10619-f002:**
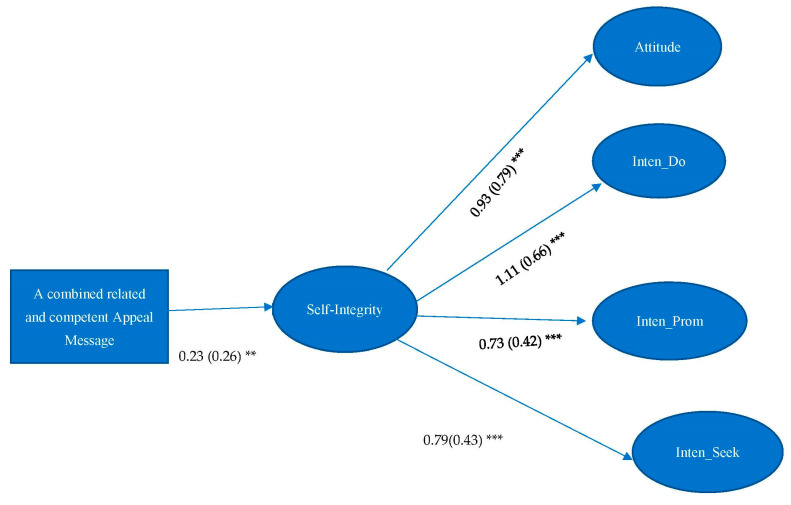
Unstandardized estimates are listed, followed by standardized estimates in parentheses. Model Fit: χ^2^ = 449.147, df = 224, *p* < 0.001; RMSEA < 0.001, 90% C.I.: 0.064–0.084, CFI = 0.920, SRMR = 0.057. Note: ** *p* < 0.01. *** *p* < 0.001. Attitude = Attitude toward organ donation, Inten_Do = Intention of becoming organ donors, Inten_Prom = Intention of promoting organ donation, Inten_Seek = Intention of seeking organ donation-related information.

**Figure 3 ijerph-19-10619-f003:**
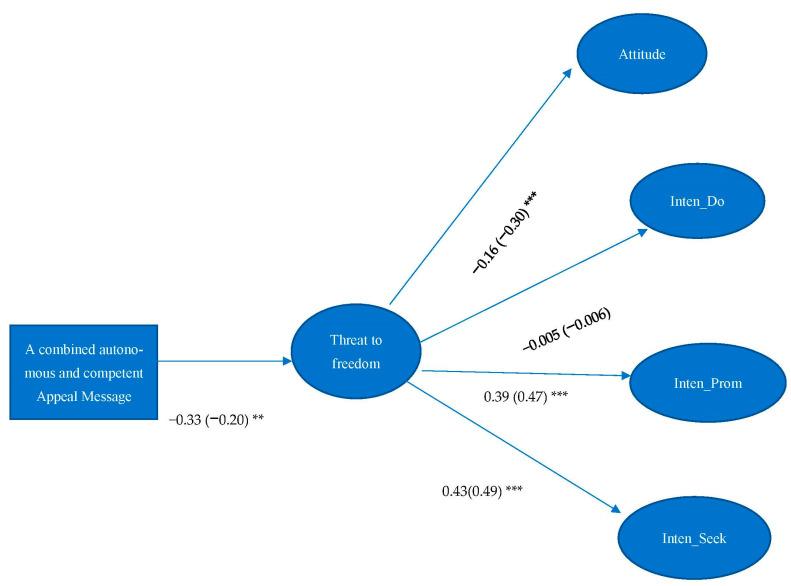
Unstandardized estimates are listed, followed by standardized estimates in parentheses. Model Fit: χ^2^ = 477.828, df = 224, *p* < 0.001; RMSEA < 0.001, 90% C.I.: 0.069–0.089, CFI = 0.920, SRMR = 0.072. Note: ** *p* < 0.01. *** *p* < 0.001. Attitude = Attitude toward organ donation, Inten_Do = Intention of becoming organ donors, Inten_Prom = Intention of promoting organ donation, Inten_Seek = Intention of seeking organ donation-related information.

**Figure 4 ijerph-19-10619-f004:**
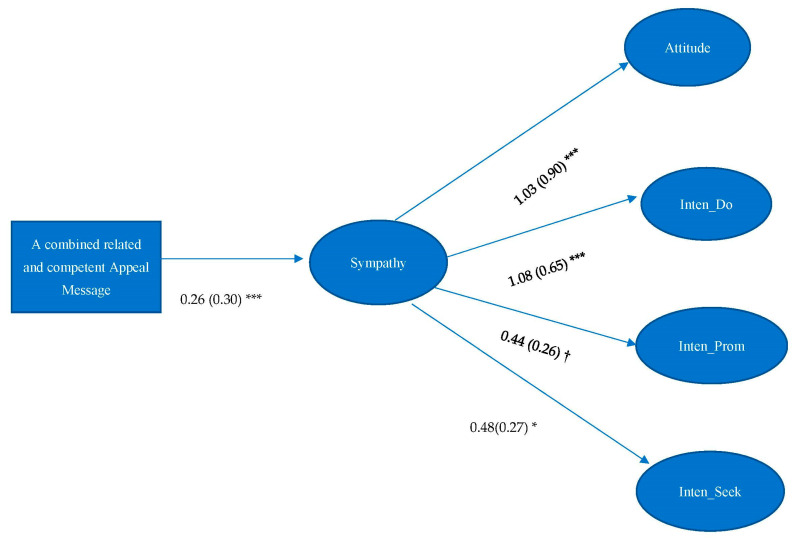
Unstandardized estimates are listed, followed by standardized estimates in parentheses. Model Fit: χ^2^ = 420.616, df = 202, *p* < 0.001; RMSEA < 0.001, 90% C.I.: 0.067–0.088, CFI = 0.915, SRMR = 0.071. Note: † *p* < 0.10. * *p* < 0.05. *** *p* < 0.001. Attitude = Attitude toward organ donation, Inten_Do = Intention of becoming organ donors, Inten_Prom = Intention of promoting organ donation, Inten_Seek = Intention of seeking organ donation-related information.

**Figure 5 ijerph-19-10619-f005:**
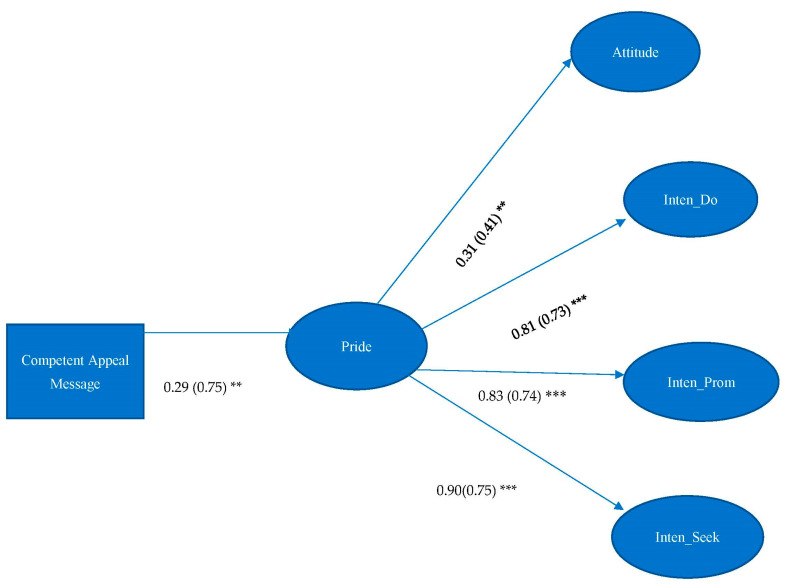
Unstandardized estimates are listed, followed by standardized estimates in parentheses. Model Fit: χ^2^ = 383.626, df = 202, *p* < 0.001; RMSEA < 0.001, 90% C.I.: 0.060–0.081, CFI = 0.934, SRMR = 0.058. Note: ** *p* < 0.01. *** *p* < 0.001. Attitude = Attitude toward organ donation, Inten_Do = Intention of becoming organ donors, Inten_Prom = Intention of promoting organ donation, Inten_Seek = Intention of seeking organ donation-related information.

## Data Availability

The data presented in this study are available on request from the corresponding author.

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
