# Peer review of "Examining the Effect of Self-Determined Appeal Organ Donation Messages and Respective Underlying Mechanism"

_ijerph, 2022, doi:10.3390/ijerph191710619_

Round 1

Reviewer 1 Report

This paper looks at the motivations and influences of people's attitudes and intentions toward organ donation, which is a fascinating topic. It offers a good frame to consider this issue. However, terms like the threat to freedom, self-integrity, pride, and sympathy, are very too general when it comes to particular behaviour like individual organ donation. Some issues need to be addressed explicitly: how are these terms applied in the readings or questionaries? How do the attitudes reflected in the self-reported data result from reading messages instead of participants' daily experiences (education, cultural background etc.)? It would be risky to link the attitude of individuals and some controlled reading materials. In addition, the author needs to write a more comprehensive literature review. Countless references are listed in the end, but nothing much can tell in the article. What kind of methods were used in the related research? How do they reach the conclusions? Organ donation is just one of the behaviours. There is much influential related research in the behaviour research field. Some should be included. 

Reviewer 2 Report

The article submitted to me for review is a good example of a scientific paper

The basic problem of the article presented to me was to answer the question how intrinsic motivation and its respective underlying mechanism influence people’s attitude and intentions of organ donation. The Author / s made insightful hypotheses, based on reports of previous studies, which methodologically correctly solved the problem.

The issue of organ donation is a challenge on the medical front. The issue of promoting human organ donation is a matter of psychology. Hence, the topic taken up by the Author/s is a place of good cooperation between psychology and medicine. As the Author/s themselves point out, the article is a response to the existing knowledge gap in the literature in this area.

The article approaches the issue of organ donation from a psychological perspective. The Author/s consider what kind of information about donation will most effectively encourage people to join in.

I would consider, and propose, a more thorough reference of the authors' own research results obtained to the theory of mastery of trepidation and reactance. The existing explanation is too general.

The results obtained and their interpretation consistently answer the hypotheses.

References, tables, figures are made in accordance with existing APA standards.

Round 2

Reviewer 1 Report

The manuscript has been sufficiently improved this time. The methods and arguments make more sense.